# Sapienic Acid Metabolism Influences Membrane Plasticity and Protein Signaling in Breast Cancer Cell Lines

**DOI:** 10.3390/cells11020225

**Published:** 2022-01-11

**Authors:** Ertan Küçüksayan, Anna Sansone, Chryssostomos Chatgilialoglu, Tomris Ozben, Demet Tekeli, Günel Talibova, Carla Ferreri

**Affiliations:** 1Department of Biochemistry, Faculty of Medicine, Alanya Alaaddin Keykubat University, Antalya 07070, Turkey; ertankucuksayan@gmail.com; 2ISOF, Consiglio Nazionale delle Ricerche, 40129 Bologna, Italy; anna.sansone@isof.cnr.it (A.S.); chrys@isof.cnr.it (C.C.); 3Department of Biochemistry, Faculty of Medicine, Akdeniz University, Antalya 07070, Turkey; ozben@akdeniz.edu.tr; 4Department of Pediatric Hematology-Oncology, Faculty of Medicine, Akdeniz University, Antalya 07070, Turkey; dmet_cskn@hotmail.com; 5Department of Histology and Embryology, Faculty of Medicine, Akdeniz University, Antalya 07070, Turkey; guneltalibova777@gmail.com

**Keywords:** membrane fatty acids, breast cancer, membrane remodeling, fatty acid supplementation, cancer signaling activation, membrane lipidome

## Abstract

The importance of sapienic acid (6c-16:1), a monounsaturated fatty acid of the n-10 family formed from palmitic acid by delta-6 desaturase, and of its metabolism to 8c-18:1 and sebaleic acid (5c,8c-18:2) has been recently assessed in cancer. Data are lacking on the association between signaling cascades and exposure to sapienic acid comparing cell lines of the same cancer type. We used 50 μM sapienic acid supplementation, a non-toxic concentration, to cultivate MCF-7 and 2 triple-negative breast cancer cells (TNBC), MDA-MB-231 and BT-20. We followed up for three hours regarding membrane fatty acid remodeling by fatty acid-based membrane lipidome analysis and expression/phosphorylation of EGFR (epithelial growth factor receptor), mTOR (mammalian target of rapamycin) and AKT (protein kinase B) by Western blotting as an oncogenic signaling cascade. Results evidenced consistent differences among the three cell lines in the metabolism of n-10 fatty acids and signaling. Here, a new scenario is proposed for the role of sapienic acid: one based on changes in membrane composition and properties, and the other based on changes in expression/activation of growth factors and signaling cascades. This knowledge can indicate additional players and synergies in breast cancer cell metabolism, inspiring translational applications of tailored membrane lipid strategies to assist pharmacological interventions.

## 1. Introduction

Cancer metabolism and invasiveness rely on monounsaturated and polyunsaturated fatty acids (MUFA and PUFA), which play structural and functional roles, particularly as hydrophobic residues of membrane phospholipids and sustaining signaling cascades [1,2,3]. The importance of fatty acids in the cancer cell membrane lipidome has been reported in recent papers and reviews, highlighting the metabolic pathways shown in Figure 1 [3,4,5,6], i.e., the formation of the MUFA oleic acid from the saturated fatty acids (SFA) palmitic and stearic acids. Moreover, in breast cancer, which has surpassed lung cancer as the most commonly diagnosed cancer [7], fatty acid biomarkers have been studied and have been found to give important clinical indications [4]. On the other hand, since eukaryotic cells, and among them, cancer cells, cannot be formed without PUFA, the two families of n-6 and n-3 fatty acids are needed, starting from the dietary intakes of essential fatty acid (EFA) precursors, linoleic and alpha-linolenic acids, respectively [3] (Figure 2). More recently, palmitic acid was found to be involved by an unusual pathway of delta-6 desaturation and transformed into sapienic acid (SA) (Figure 1) [8,9]. Indeed, the presence of SA was first evidenced by us in lipid fractions isolated from human plasma [8], and its significantly different levels were found in erythrocyte membrane phospholipids and plasma cholesteryl esters when comparing an obese cohort to healthy subjects [9].

It is worth noting that the SA transformation is considered a minor path since palmitic acid is mostly transformed by elongase or delta-9 desaturase, as shown in Figure 1, and also because delta-6 desaturase is known to work mainly with PUFA (Figure 2). On the other hand, it is known that SA biosynthesis occurs in sebocytes and, according to Figure 1, proceeds with the formation of the elongation product 8cis-18:1 and the PUFA sebaleic acid [10]. The three fatty acids in Figure 1 (right side) represent the n-10 fatty acid family, which are positional isomers of the n-9 and n-7 MUFA, i.e., oleic and palmitoleic acids shown in Figure 1, and of the n-6 PUFA linoleic acid shown in Figure 2 [11,12]. After our first detection and quantitation of SA in human blood [8], the “systemic” presence of sapienic acid in membranes of blood cells was reported in phagocytic cells both of human and murine origins, testing its anti-inflammatory activity but at a higher dose (25 µM) than the one (10 µM) known for MUFA, such as oleic and palmitoleic acids [13]. Our objective was to detect the impact of such pathways on human diseases, particularly in cancer. Indeed, the desaturase enzyme known as stearoyl-CoA desaturase (SCD-1), carrying out the transformation of stearic acid to oleic acid (Figure 1), is crucial for cancer development and invasiveness [14,15]. This finding led to specific desaturase inhibition strategies to cause detrimental effects in cancer cell proliferation [16].

In Caco-2 cell lines, we demonstrated the occurrence of the full metabolism of SA to sebaleic acid with a distinctive effect on membrane fluidity features detected by general polarization measurements [11]. Further, we followed up the n-10 fatty acids in prostate adenocarcinoma cell lines and found interesting differences in this metabolism in more aggressive cancer cell types [17]. The expression of delta-6 desaturase and its ability to form SA was then evidenced in different cancer cell types, demonstrating that proliferation is connected with SA supplementation when this enzyme is inhibited [18]. On the other hand, sebaleic acid metabolism was demonstrated in sebocytes via enzymatic oxidation that produces 5-hydroxy-(6*E*,8*Z*)-octadecadienoic acid (5-HODE), 5-oxo-(6*E*,8*Z*)-octadecadienoic acid (5-oxo-ODE), 5*S*,18-dihydroxy-(6*E*,8*Z*)-octadecadienoic acid, and 5-oxo-18-hydroxy-(6*E*,8*Z*)-octadecadienoic acid, with activity on neutrophil infiltration and induction of inflammatory responses [19].

In cancer cells, no data are available on signaling cascades derived from n-10 fatty acids. We found it intriguing that a strong expression of the epidermal growth factor receptor (EGFR, also known as ERBB1 or HER1), belonging to a family of tyrosine kinase receptors, was reported in undifferentiated sebocytes at the periphery of mouse and human sebaceous glands [20]. Further studies concerning the effects of n-10 FA on signaling cascades in the cellular environment, including cancer cells, are lacking.

Pursuing our interest in n-10 fatty acids, here we present the results of SA supplementation in breast cancer cells, studied in terms of membrane FA incorporation and remodeling, and associating for the first time the oncogenic signaling cascade of EGFR, AKT (protein kinase B) and mTOR (mammalian target of rapamycin) [21]. We used 50 µM SA concentration in 3 different breast cancer cell lines: MCF-7 as the hormone-dependent cell line (both estrogen- and progesterone-receptor positive, ER and PR), and MDA-MB-231 and BT-20 as triple-negative breast cancer cells (TNBC). Membrane fatty acid remodeling, expression and phosphorylation (activation) of EGFR, AKT and mTOR were followed up in the first three hours and compared to untreated cells. The scope of this study is to gather first indications in breast cancer cell lines that SA and its metabolism to n-10 MUFA and PUFA can affect both membrane composition due to phospholipid remodeling and the expression/activation of cell signaling, expanding knowledge on fatty acid-based mechanisms that can be involved in cancer invasiveness and escape.

## 2. Materials and Methods

Cis and trans FAME, dimethyl disulfide, iodine, cholesterol, sphingomyelin were purchased from Merck-Sigma-Aldrich (Darmstadt, Germany) and used without further purification; sapienic acid methyl ester, 8cis-18:1 methyl ester and sebaleic acid methyl ester were purchased from Lipidox (Lidingö, Sweden). The solvents chloroform, methanol, isopropanol, diethyl ether and n-hexane (HPLC grade) were purchased from Carlo Erba (Milan, Italy) and used without further purification. POPC (1-palmitoyl-2-oleoyl-sn-glycero-3-phosphocholine), POPE (1-palmitoyl-2-oleoyl-sn-glycero-3-phosphoethanolamine) and POPS (1-palmitoyl-2-oleoyl-sn-glycero-3-phosphoserine) were purchased from Larodan (Solna, Sweden) and used without further purification. Silica gel analytical thin-layer chromatography (TLC) was performed on Merck silica gel 60 plates of 0.25 mm thickness (Merck, Darmstadt, Germany) and spots were detected by spraying the plate with a cerium ammonium sulfate/ammonium molybdate reagent. Phospholipid classes were analyzed by HPLC (Agilent 1200, Santa Clara, CA, USA) equipped with an RP18 column (EC 150/4.6, Nucleodur C18 HTEC, 5 µM Macherey-Nagel, Düren, Germany) using the following isocratic condition: buffer ammonium formate 1mM/ MeOH/ 2-propanol = 10/20/70, detector UV 203. Phospholipids were identified by comparison with the retention times of commercially available references. Values are expressed in relative quantitative percentages (% rel. quant.), i.e., are obtained in µg/mL by calibration curves and converted to percentages of each lipid class over the total amount of lipid classes taken as 100%. The data are shown in Appendix A together with a description of calibrations and measurement procedures and parameters, and with a representative HPLC chromatogram (Appendix A) in Appendix A.

Gas chromatography and detection with a flame ionization detector (GC-FID) were carried out by Agilent 6850 equipment (Agilent, Milan, Italy) in splitless mode using a 60 m × 0.25 mm × 0.25 µm (50%-cyanopropyl)-methylpolysiloxane column DB23 and the following oven program: temperature started from 165 °C, held for 3 min, followed by an increase of 1 °C/min up to 195 °C, held for 40 min, followed by a second increase of 10 °C/min up to 240 °C, and held for 10 min. A constant pressure mode (29 psi) was chosen with helium as the carrier gas. Fatty acid methyl esters (FAME) obtained from the membrane phospholipids, as described in Section 2.5, were separated and quantified using previously published procedures and commercially available standard references [9,11,12,17]. A representative GC run is shown in Appendix A. FAME are expressed as relative percentages (mean ± SD) of the quantitative values of each fatty acid obtained by calibration curves calculated directly by the software Chem Station Agilent of GC instrument. The procedures for LOD, LOQ and calibration of GC instrument have been reported previously [9,11,12,17]. FAME data are reported in Appendix A and are shown as graphs in Figure 1 in the text.

### 2.1. Cell Culture and Treatment

MCF-7, MDA-MB-231 and BT-20 were purchased from American Type Culture Collection (ATCC, Manassas, VA, USA). MCF-7, MDA-MB-231, and BT-20 were grown in Dulbecco’s Modified Eagle’s medium (DMEM), in Roswell Park Memorial Institute (RPMI) 1640, and in Eagle’s minimal essential medium (EMEM), respectively. Cells were supplemented with fetal bovine serum (FBS) (Sigma, Darmstadt, Germany), 1% l-glutamine (Sigma, Germany) and 1% penicillin/streptomycin (Sigma, Darmstadt, Germany), and incubated at 5% carbon dioxide and 37 °C in a humidified atmosphere. A day after seeding, the culture medium was refreshed. A fresh sapienic acid batch was prepared in 100% DMSO and used at 50 µM concentration for all the cultures, taking care that the final concentration of DMSO in the culture medium never exceeded 0.01% (*v*/*v*). The same DMSO concentration was present in control experiments. Incubations were left for the established times. In the “pulse and chase” experiment, used as control of the immediate effect of SA (CTR-PC), the addition was carried out, and cells were kept in culture for 2–3 min, then washed and used for further determination as described in the following sections.

### 2.2. Cell Proliferation Assay

Cell proliferation/viability were determined using MTT assay [22], as done previously [23]. Briefly, cells were seeded in 96-well plates at 10,000 cells/well. A day after seeding, the culture medium was changed, and SA was incubated at different doses (0–500 µM) for 24 h, then incubated with MTT for 3 h. The absorbance at 570 nm was determined by a microplate reader (BioTek, Instruments Inc., Winooski, VT, USA). To determine the viable cells, the absorbance values were compared with the control group and calculated as the percentage of viable cells. Viability analyses were carried out in SA-incubated cells in order to calculate the IC50 for the three breast cancer cells.

### 2.3. Western Blotting

After 24 h incubation of the three breast cancer cells at 37 °C under the experimental conditions described in Section 2.1, the assay was carried out following previously reported procedures [23,24]. Total proteins were extracted from each experiment with radioimmunoprecipitation assay lysis buffer (Pierce; Thermo Fisher Scientific, Inc. Waltham, MA, USA). Protein samples were then quantified with a Bradford Protein Assay kit (Pierce; Thermo Fisher Scientific, Inc., Waltham, MA, USA). Proteins (30 μg/pore) were separated by 10% sodium dodecyl sulphate-polyacrylamide (SDS-PAGE) gel and transferred onto polyvinylidene difluoride membrane (PVDF membrane, Millipore, Darmstadt, Germany) using wet electroblotting. The membrane was then blocked overnight with skimmed milk at 4 °C and incubated with antibodies against phospho-EGF receptor (Tyr1068) (p-EGFR; 1:1000; cat. no. 3777), EGFR (1:1000; cat. no. 4267), phospho-AKT (Ser473) (p-AKT; 1:1000; cat. no. 4060), Akt (1:1000; cat. no. 4691), phospho-mTOR (Ser2448) (p-mTOR; 1:1000; cat. no. 5536), mTOR (1:1000; cat. no. 2983), and beta-actin (1:10,000, cat. no. 5174) All primary antibodies were purchased from Cell Signaling Technology (CST, Leiden, The Netherlands). After washing with tris-buffered saline with Tween 20 (TBST), the membrane was incubated with horseradish peroxidase-conjugated secondary antibodies (1:10,000; cat. no. 5450-0010; Sera Care, Gaithersburg, MD 20878, USA) and visualized with an enhanced Luminata Forte Western HRP Substrate (Millipore, Darmstadt, Germany). Beta-actin acted as an internal control and was quantified using the G:BOX Chemi XRQ system (Syngene, Cambridge, UK). The data of all experiments are shown in Appendix A and Figure 2 in the text.

### 2.4. Preparation of Cells for Phospholipid Pellets

Petri dishes containing 3 × 10^6^ cells were incubated in the conditions described in Section 2.1 with 50 µM SA for the established times. Cells were detached using accutase, thoroughly washed with phosphate buffer saline (PBS), and pellets were obtained as described in the following Section 2.5.

### 2.5. Lipid Extraction and Fatty Acid Analysis of MCF-7, MDA-MB-231 and BT-20 Cell Lines

A sample (3 × 10^6^ cells) of MCF-7, MDA-MB-231 or BT 20 (controls and supplemented with sapienic acid) suspended in 1 mL PBS was added to 0.5 mL triple-distilled water and centrifuged at 14,000 rpm for 5 min at 4 °C to obtain a membrane pellet. The membrane pellet was reconstituted in triple-distilled water (1 mL), and 20 µL were injected in the HPLC equipment, using the conditions described above, to determine lipid classes and free cholesterol (see Appendix A—section “HPLC detection of lipid classes” for details of calibration and protocol procedures. Data are reported in Appendix A is a representative HPLC run). The remaining membrane pellet underwent lipid extraction with 2:1 chloroform:methanol (4 × 4 mL) according to the Folch method [25]. The organic layers, dried on anhydrous Na_2_SO_4_, were evaporated to dryness, and the total lipid content (0.6–0.85 mg in each sample) was checked by TLC (n-hexane:diethyl ether 9:1). To examine the fatty acid composition of the membrane pellet, extracts were converted into FAME (fatty acid methyl esters) by adding 0.5 M KOH in MeOH (0.5 mL), stirring the reaction for 10 min and quenching by brine (0.5 mL). FAME were extracted with n-hexane (4 × 2 mL), dried on anhydrous Na_2_SO_4_, evaporated to dryness, dissolved in n-hexane (10 µL) and analyzed by GC using the standard references for peak identification and quantitation as already described [9,11,12,17]. Appendix A and Figure 1 in the text show the results of the fatty acid analyses.

### 2.6. Statistical Analysis

Results are given as mean ± SD (deviation standard). Statistical analysis was performed using GraphPad Prism 8.0 software (GraphPad Software. Inc., San Diego, CA, USA). To compare results, a nonparametric unpaired two-tailed *t*-test was used with a 95% confidence interval. Correlations between protein activation and membrane fatty acids were performed using Spearman correlation and visualized by heat map. Significance was given for *p* < 0.05.

## 3. Results

### 3.1. Cell Cultures and Sapienic Acid Supplementation

Cultured MCF-7, BT-20, and MDA-MB-231 cells, in the conditions described in Section 2, were treated with 50 μM sapienic acid for 24 h at 37 °C, analyzing in the first 3 h the expression and activation of EGFR, mTOR and AKT signaling in combination with the analysis of the membrane fatty acid patterns. In order to decide about the SA dosage, dose-dependent cytotoxicity at various concentrations of SA was determined in MCF-7, MDA-MB-231, and BT-20 cells by MTT assay as described in Section 2. The IC50 values of SA were 128.9 μM, 147.8 μM, and 162.3 μM for MCF-7, MDA-MB-231 and BT-20 cells for 24 h, respectively, demonstrating that the IC50 concentrations for TNBC cells were greater than in MCF-7 cells. In addition, the IC50 concentration of SA was higher in BT-20 cells than MDA-MB-231 cells at 24 h. We followed cell viability over the 24 h and at different concentrations of SA, finding that BT-20 cells were the most resistant to high concentrations compared to MCF-7 and MDA-MB-231 cells (data not shown). We did not further investigate the viability of the cell lines. These data were mainly used to establish the SA concentration for the experiments. We decided to use 50 μM SA in the 3 cell lines since the effects of sapienic acid were previously tested at 10 and 25 µM concentrations [13] in normal cells (phagocytic cells). We thought that in cancer cells, the 50 µM concentration could simulate increased production of this compound for the specific metabolism (delta-6 desaturase activation). However, it is still comfortable with the IC50 since it is 2–3 times lower than the IC50 for all cell types tested in this work. Moreover, we demonstrated that no lipid accumulation (production of triglycerides) was obtained at this concentration, and at this concentration, it should trigger its metabolism to the other n-10 members.

### 3.2. Membrane Fatty Acid Remodeling under Sapienic Acid Supplementation in Three Breast Cancer Cells

We were interested in monitoring the membrane fatty acid contents influenced by SA supplementation. We performed follow ups at 0.5 h, 1 h, 2 h and 3 h after supplementation in triplicated cell culture experiments, comparing with control cells in the same conditions without SA supplementation. Each sample for membrane lipidome analysis, containing a counted cell number of 3 × 10^6^ cells, followed the procedure described in Section 2 for phospholipid isolation, work-up for fatty acid methyl ester (FAME) formation and detection using gas chromatography (GC), as previously carried out in Caco-2 cell lines [11]. Fatty acids belonging to membrane phosphoglyceryl esters of the 3 cell lines are shown in Appendix A, as mean ± SD, corresponding to >97% of the total peaks detected in the GC analysis (see Appendix A for a representative GC run). Each fatty acid was calculated quantitatively based on calibration curves and reported as percentages of the total fatty acid recognized by appropriate standards (% rel. quant.).

In Figure 1, the data of the main statistically significant FAs of the three cell lines are reported as a function of the time (0–3 h) in order to have a snapshot of the changes for (A) MCF-7; (B) MDA-MB-231 and (C) BT-20 cells (changes are reported as % of controls). It is worth noting that control cells showed a different distribution among SFA, MUFA and PUFA residues in membrane lipidomes which were different for each cell line, although they were all of breast cancer type. SA supplementation induced different trends of membrane fatty acid changes in the three cell lines, as follows:(a)In MCF-7 cells (Appendix A and Figure 1A), the rapid incorporation of SA (0.5 h) brought the increase of its levels 10 times compared to controls after 3 h (from 0.8% to 8.48%) (*p* ≤ 0.0009); the sapienic/palmitoleic ratio after 3 h changed accordingly (*p* = ≤ 0.0009). As shown in Figure 1A, the incorporation of 8cis-18:1 in membrane phospholipids, implying the elongation of SA occurring in the cells, was immediate, bringing a significant increment of total n-10 fatty acids from 0.5 to 3 h (*p* ≤ 0.04 to 0.007). In this type of breast cancer cell, the level of the PUFA sebaleic acid in membrane lipids remained untouched compared to controls. Membrane remodeling included diminution of some MUFA levels, particularly oleic and vaccenic acids (Figure 1A), and PUFAs, such as n-3 DHA and EPA, n-6 arachidonic acid, especially at 1 h, and DGLA, but without significance;(b)In MDA-MB-231 (Appendix A and Figure 1B), the incorporation of SA in membrane fatty acids was also immediate at 0.5 h, but then remained almost constant up to 3 h (*p* ≤ 0.0009), together with elongation/incorporation of 8cis-18:1 also occurring since 0.5 h and significant at 3 h (*p* ≤ 0.0009) and a low formation/incorporation of PUFA sebaleic acid in the first 2 h, which was not significant. The total content of n-10 fatty acids significantly increased at all incubation times with respect to controls. The sapienic/palmitoleic acid ratio was significantly increased after 30 min and 3 h. In the meantime, a profound and significant remodeling involved all the fatty acid families, as shown in Figure 1B: (i) SFA (palmitic acid C16:0, stearic acid C18:0 and C20:0) decreased; (ii) the MUFA oleic acid increased (*p* ≤ 0.0009); (iii) for n-6 PUFA (Figure 1B, right graph), the decrease of arachidonic acid and DGLA (*p* ≤ 0.04; 0.007; *p* ≤ 0.0009) was accompanied by the increase of linoleic acid up to the first 2 h (*p* ≤ 0.0009); for n-3 PUFA, after 2 h of incubation, both 22:5 DPA (*p* ≤ 0.04) and 22:6 DHA (*p* ≤ 0.04) diminished with a recovery of both levels after 3 h;(c)In BT-20 (Appendix A and Figure 1C), SA was significantly incorporated, also after elongation to 8cis 18:1, during the 3 h incubation (*p* ≤ 0.0006) and, only in this cell line, the level of sebaleic acid in membrane phospholipids significantly increased after 30 min (*p* ≤ 0.007); the content of total n-10 fatty acids increased in the time and doubled significantly after 2 h (*p* ≤ 0.0006), becoming almost 4 times more respect to controls after 3 h of incubation (*p* ≤ 0.0006). The sapienic/palmitoleic acid ratio significantly changed after 2 and 3 h of incubation (*p* ≤ 0.0006), reaching the highest value in BT-20 cells compared to the other 2 cell lines. In this cell line, the remodeling involved: (i) the diminution of total SFA and, in particular, palmitic acid (*p* ≤ 0.0006); (ii) the MUFA oleic and vaccenic acids decrease (*p* ≤ 0.007 and *p* ≤ 0.0006); in the PUFA family, we note the increase of n-6 DGLA and ARA in the first 30 min (*p* ≤ 0.007; *p* ≤ 0.0006), together with the significant increase of the n-3 fatty acids, in particular EPA (*p* ≤ 0.04 at 3 h).

We also quantified by HPLC, using calibration with standard references as described in Section 2 and in Appendix A, the main lipid classes of cell membranes, in particular phosphatidylcholine (PC), phosphatidylserine (PS), sphingomyelins (SM) and cholesterol (Appendix A), which represent almost 90% of the total lipid classes in breast cancer cell lines [26,27]. A representative HPLC run is shown in Appendix A. Differences among the three cell lines were evidenced, as follows: (a) in MCF-7, a significant decrease of sphingomyelins (SM) was shown after 2 h (*p* ≤ 0.05); (b) in MDA-MB-231, a decrease of cholesterol (CHO) after 3 h (*p* ≤ 0.05) and phosphatidylcholine (PC) was seen after 2 h (*p* ≤ 0.05); (c) in BT-20 a significant increase of SM was seen after 1 h (*p* ≤ 0.03) and 3 h (*p* ≤ 0.004).

Finally, we also performed pulse-and-chase experiments at 50 μM SA in the 3 cell lines and compared the effects of membrane fatty acid remodelling with control cells at a short exposure time of 2–3 min (CTR-PC). The results are shown in Appendix A. MCF-7 did not show any significant changes in fatty acid composition. Instead, MDA-MB-231 responded immediately with variations of n-10 FA, incorporating SA and its elongated metabolites, including the endogenous PUFA, sebaleic acid. This incorporation was accompanied by remodeling with specific diminution of di-homo gamma-linolenic acid (C20:3 n-6). Lastly, the brief exposition for BT-20 cells did not show significant n-10 incorporation but responded immediately with the remodeling of SFA and PUFA residues, decreasing the former and increasing the latter in the n-6 (ARA) and the n-3 (EPA and DPA) series.

### 3.3. Expression and Phosphorylation of EGFR, AKT and mTOR in Breast Cancer Cells Treated with Sapienic Acid

During incubation of MCF-7, MDA-MB-231, and BT-20 cells with 50 μM sapienic acid, the expression of the proteins EGFR, AKT and mTOR and their activation by phosphorylation (p-EGFR, p-AKT, p-mTOR) were detected by Western blotting. Appendix A show the results of the three cell lines regarding the three proteins. Figure 2 shows graphics of the amounts of phosphorylated proteins at 0, 1, 2, 3 h, as well as in the pulse and chase experiment (CTR-PC). In MCF-7 cells, treatment with 50 μM SA induced EGFR expression and increased its expression approximately 3-fold compared to control for 2 h (Appendix A), whereas there was a significant decrease in EGFR activation from the first hour (Figure 2A, left graph). On the other hand, EGFR activation was already very high in MDA-MB-231 cells, a TNBC cell (Appendix A), and there was a significant increase in EGFR activation up to 3 h with SA treatment (Figure 2A, middle graph). In BT-20 cells, another TNBC cell, there was a strong EGFR expression in controls, which was also maintained for 2 h (Appendix A), together with a significant increase in EGFR activation for 3 h with SA treatment (Figure 2A, right graph), as observed in MDA-MB-231 cells. In the evaluation of mTOR in these cell lines, the expression increased in the MCF-7 cell line after the SA supplementation, at least up to 2 h (Appendix A), and the activation was also increased up to 3 h (Figure 2B, left graph).

For the MDA-MB-231 cell line, mTOR expression diminished compared to control cells (Appendix A), whereas mTOR activation steadily increased over the 3 h (Figure 2B, middle graph). In the BT-20 cell line, we found an increase of mTOR activation with SA treatment in 1 h (Appendix A) accompanied by strong activation (Figure 2B, right graph). In evaluating AKT protein in the three cell lines, we observed the expression only in the MCF-7 cell line in the first hour (Appendix A), but the activation increased up to 3 h (Figure 2C, left graph). In MDA-MB-231 cells, we found that SA treatment increased AKT expression (Appendix A). Activation approximately increased 3-fold in the first and second hours and then decreased in the third hour compared to the controls (Figure 2C, middle graph). In BT-20 cells, AKT expression was maintained only in the first 1–2 h (Appendix A), whereas AKT activation was significantly increased by SA treatment along the 3 h (Figure 2C, right graph).

Follow-up of the instantaneous effect of SA on the signaling proteins, when it comes in contact with cells for a few minutes (CTR-PC experiment), gives the extent of the sensitivity in the three cell lines (Figure 2), mainly regarding the immediate membrane remodeling: MCF-7 and MDA-MB-231 cell lines showed mTOR and AKT activation while BT-20 showed only a significant diminution of AKT at this short time incubation (Figure 2).

We finally run a statistical evaluation of the data in our hands, particularly regarding correlations between the n-10 FA levels and the activation/expression of signaling proteins, to obtain a first map of the significant effects induced by the presence of n-10 FA in the membrane phospholipids. As shown in Figure 3, 50 µM SA supplementation produced different results in each cell line. In particular, positive correlations were found as follows: (a) in MCF-7 cells, sapienic and sebaleic acids and the total n-10 FA correlated positively with p-AKT and p-mTOR (*p* < 0.005 and *p* < 0.05, respectively); (b) in MDA-MB-231 cells, only total n-10 FA levels correlated positively with p-mTOR (*p* < 0.05); (c) in BT-20 cells, total n-10 FA levels were positively correlated with p-EGFR. It is worth recalling that BT-20 showed the highest IC50, and this is the only cell line to have EGFR activation correlated with the n-10 FA levels.

## 4. Discussion

In the present study, we were interested in the effects of SA in ER-positive (MCF-7) and -negative (MDA-MB-231 and BT-20) breast cancer cells in connection with the activation of EGFR expression and the AKT/mTOR pathway. In our experiments, we chose to follow up the time window of the first 3 h since the effects of fatty acid supplementation have previously been shown to be connected to the fast membrane lipidome remodeling sustained by the well-known Lands’ cycle [28]. Using a neuroblastoma cell line, we carried out pulse-and-chase (PC) experiments. After a short time of fatty acid supplementation, cells were washed and suspended in fresh medium to continue the culture. In these conditions, we demonstrated that fatty acids induce immediate changes to membrane composition and that, compared with cell cultures exposed to fatty acid supplementation for an extended period, the PC cell fate is different [28].

In the present study, for the first time, membrane remodeling is followed up together with EGFR, AKT and mTOR protein expressions and activation during SA supplementation in three breast cancer cell lines (MCF-7, MDA-MB-231, BT-20) chosen for their different characteristics of hormone responders and resistance. The incubation with 50 μM SA was chosen since it did not affect cell viability. In literature, effects of sapienic acid were reported in normal cells (phagocytic cells) at 10 and 25 µM concentrations, indicating that only the latter has an anti-inflammatory effect [13]. We decided to use the 50 µM concentration, considering that our experiment is in cancer cells. We wanted to simulate a more significant production of this compound in cancer cells (delta-6 desaturase activation), which is still comfortable with the IC50 (2–3 times higher concentrations in all cell types tested in this work; see Section 3). Moreover, we demonstrated that no lipid accumulation (formation of triglycerides) is obtained at this concentration, and transformation into its n-10 congeners and incorporation in membrane lipids occur, as shown in Figure 1.

Under such supplementation, the first relevant result of our work is that the three cancer cell lines did not have the same behaviour, showing important differences in both membrane fatty acid remodeling and expression/activation of signaling proteins (cf., Figure 1 and Figure 2).

In MCF-7 cells, a decrease of EGFR activation was seen, but at the same time, p-mTOR and p-AKT were increased (Figure 2A–C, left graphs); since increased levels of sapienic acid and total n-10 fatty acids were found in the membrane lipidome of this cell line, a “direct” effect of the membrane asset can be connected with the propagation of this signaling pathway. This was also confirmed by the correlation between fatty acids and signaling in Figure 3A; among the three cell lines, MCF-7 was the one containing the highest level of SFA (up to 70% of the total FA), whereas the other cell lines reached 50% SFA over the total FA content. Since FA contents can vary according to the growth conditions, we attributed these changes to the specific experimental conditions rather than a stable difference due to the cancer types. On the other hand, culture conditions influence membrane lipidome and fatty acid compositions, as firstly evidenced in neuroblastoma cell lines [28], and requests for more attention for such an issue have been detailed by other authors [29]. In our conditions, the MCF-7 cell line had strikingly different behavior in the membrane fatty acid remodeling compared to the other two TNBC cell lines: the incorporation of n-10 FA was accompanied by the incorporation of oleic acid, whereas PUFA moieties were involved, but not significantly (Figure 2A). We recall that the enrichment of oleic acid in membrane phospholipids is an important biomarker of SCD-1 activation and tumor proliferation, whereas SCD-1 inhibition is obtained by exogenous supplementation of oleic acid [30]. Desaturase activity with MUFA increases, but without considering n-10 FA and their effects, has been connected with the role of membrane fluidity in cancer to directly activate the AKT cascade [14,15]. The formerly suggested role of membrane plasticity and desaturase enzymes in the formation of unsaturated fatty acid moieties [14,15,31] has been recently reaffirmed [32,33].

In the case of the two TNBC cell lines, membrane lipidome and protein signaling changed by SA supplementation and in a distinctive manner for each cell line, as follows: in MDA-MB-231 cells, EGFR activation occurred at 3 h, whereas the mTOR pathway responded immediately and the p-AKT increase occurred in the first 2 h (Figure 2A–C, middle graphs); at the same time, in the MDA-MB-231 lipidome, oleic acid increased significantly together with n-6 linoleic acid in the first 2 h, while SFA and PUFA moieties diminished (Figure 1B); here again, the MUFA increase can have the same significance seen in MCF-7. Total n-10 levels correlate with mTOR levels, as shown in Figure 3B. In BT-20 cells, the net increase of EGRF, together with strong activation of the other signaling proteins, p-mTOR and p-AKT, were observed (Figure 2A–C, right graphs), coupled with the incorporation of the highest amount of n-10 FA among all cell lines in the 3 h monitoring (14.41 ± 1.82, see Appendix A). BT-20 was also the cell line with the highest increase in the sapienic/palmitoleic acid ratio (7 folds) compared to the other two cell lines, thus showing a specific delta-6 desaturase partition in this cell line. Only in this case, the total n-10 levels were found to be correlated with EGFR (Figure 3C). Finally, this cell line is the only one to show the net increase of n-3 EPA (Figure 1C).

Our results demonstrated that the n-10 fatty acid family is able to influence the EGFR/AKT/mTOR cascade, which is a fundamental pathway ubiquitously present in tumors to maintain growth, survival, and metastasis [34,35]. EGFR is overexpressed in more than 20% of metastatic breast cancer cases and approximately half of TNBC cases [36]. We showed that in BT-20 cell lines, there is a correlation between the total n-10 fatty acid level and EGFR expression. This result can also have an impact in anti-EGFR therapies and in cases of resistance to EGFR inhibitors, suggesting to deepen the behavior of membrane remodeling as synergies for cancer cell resistance [37]. Overexpression and activation of AKT and mTOR, as sub-pathways of EGFR, are directly linked to the development of breast cancer [38], and we demonstrated that n-10 fatty acids have an impact on these two signaling proteins, in connection with but also independently from EGFR. It should be noted that in previous studies, it was proven that EGFR activation can directly increase mTOR-mediated cell growth by forming phosphatidic acid (PA) by phospholipase D [39] and can also indirectly increase cell growth by activating mTOR via PI3K/AKT through EGFR activation [1,5]. On the other hand, activation of AKT/mTOR is known to promote tumor growth and metastasis with many mechanisms underlying the development of the disease and the resistance to AKT/mTOR inhibitors in cancer cells [40,41].

We wish to underline that examining the effects of the SA supplementation in the pulse and chase manner, mimicking a short-time exposure to an “influencer” of the cell metabolism, can help investigate how quickly consequences of SA can occur to the composition of the membrane fatty acids. The immediate transformation of SA into its metabolites, especially in the BT-20 cell line, was observed, thus suggesting that n-10 FA should be followed up in specific cancer-resistant cell lines. Additionally, we analysed lipid classes evidencing different behavior in the three cell lines. The examined lipid classes account for 90% of the species, and they have also been followed in other breast cancer cell experiments [26,27]; however, we underline the need for systematic analytic work, as reported in other studies [13].

Altogether, our results suggest that n-10 fatty acids can add important information, especially in the development of new paradigms, particularly considering the emerging role of SCD-1 desaturase and monounsaturated fatty acid (MUFA) biosynthesis in cell resistance to ferroptosis [42].

We hypothesize that the dual role of n-10 fatty acids can proceed as shown in Figure 4, i.e., (a) by direct EGFR interaction with the production of signaling cascade (in our case, mTOR and AKT). In this case, we recall the known sebaleic acid interaction with EGFR in sebocytes [20]; and (b) by the induction of membrane remodeling due to an increase in the production of unsaturated fatty acid moieties, as recently highlighted in the debate on desaturase activities and diversification of unsaturated fatty acid metabolites [43], thus influencing membrane properties, such as fluidity, known to activate AKT signaling [14,15,31]. Such mechanisms can be operative alone or synergically, depending on the specific condition, and performing a fatty acid-based membrane lipidomic analysis will be an important tool for gaining more insights on all factors influencing cancer cell growth.

This is a preliminary study, and we are aware that further work is needed in order to completely understand the scenario of membrane remodeling and membrane fluidity changes with cell signaling regarding proliferation and invasiveness. Deepening the effects of SA in the presence of different dietary fatty acid conditions can also be suggested to study reprogramming of fatty acid metabolism in cancer, as well as for therapeutical applications [44,45].

## 5. Conclusions

Our study highlights the importance of the newly discovered n-10 fatty acid family in human cancer cells, clarifying its impact for membranes in both processes of fatty acid remodeling and receptor interaction for signaling. Membranes are definitely not spectators [46], and n-10 fatty acids can act by changing both their fatty acid composition and biophysical properties, as well as by activating growth factors and signaling cascades. We believe that knowledge on fatty acid-based membrane lipidome analysis will expand the understanding of lipid metabolism and individuate related biomarkers which are applicable as translational molecular medicine tools. Because lipids are an important part of human nutrition, especially in evaluating unsaturated fatty acid intakes, integrated molecular diagnostic profiles of breast cancer patients can be envisaged as easy-to-use screening for the design of tailored dietary regimes, as is emerging in recent clinical observational studies [47,48].

## Data Availability

Data related to all experiments performed in the paper are provided in Appendix A.

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
