# Peer review of "Sapienic Acid Metabolism Influences Membrane Plasticity and Protein Signaling in Breast Cancer Cell Lines"

_cells, 2022, doi:10.3390/cells11020225_

Round 1

Reviewer 1 Report

In this paper, Küçüksayan et al. have studied changes in the lipidome of three different breast cancer cell lines when exposed to sapienic acid. In addition the authors attempted to relate the changes in the lipidome with variations in the expression/phosphorylation of EGFR, m-TOR and AKT. The authors find clear differences between the three cell lines utilized both in terms of fatty acid changes and the expression activation of the aforementioned signaling proteins. Overall, the study appears well designed and carefully carried out. This is a descriptive work, yet the results are clear and interesting and may be of pathophysiological relevance.

1) Line 21. Given that the authors are referring to cells in this sentence, the word “synthesized” would seem more appropriate than “prepared”.

2) Line 57. While the authors may be correct, it is also noteworthy that macrophages, which contain  high levels of PUFA, are recently reported to contain significant amounts of sapienic acid (JLR 59: 237, 2018).

3) Line 120. The authors indicate that in Tables S7-9 the data are expressed in µg/ml. However in the captions to these tables, it is stated: “Values are expressed as relative quantitative percentages (% rel. quant.) calculated from the quantities of lipids obtained by HPLC analysis identified and calibrated by the standard references.” Please clarify. If the statement appearing in the captions is the correct one, it would be important for the authors to explain why this way of this presenting the data was chosen. The quantification method should also be described in sufficient detail. Please also specify whether CHO refers to total cholesterol or free cholesterol in these tables. Finally, the absence of data on other relevant phospholipid classes such as phosphatidylinositol or phosphatidylglycerol is conspicuous. Was there a technical reason to justify this absence?   

4) Lines 189-193. I was very confused by these sentences. What I understood here is that the authors injected a 20-µl aliquot of a cell sample in distilled water, not a lipid extract, into the HPLC machine for lipid analysis. This makes little sense to me. Further, how does this fit with the following sentences, where the method of preparation of the lipid fraction from the remainder of the cell sample is properly described?  In line 191, I could not figure out which tables were the authors referring to.

5) Line 197. The GC-FID analyses of fatty acids appear to have been conducted carefully and I have no concerns about these. The authors show good data on the phospholipid fatty acid profiles of their three cell lines under study (Tables 1-6). A fatty acid analysis of the different phospholipid subclasses separately (PC, PE, PS) would have also been very informative.

6) Discussion – general. This section is too long and at times repetitive with Results. I suggest the authors to shorten this section and make it more focused on the results at hand.

Author Response

In this paper, Küçüksayan et al. have studied changes in the lipidome of three different breast cancer cell lines when exposed to sapienic acid. In addition the authors attempted to relate the changes in the lipidome with variations in the expression/phosphorylation of EGFR, m-TOR and AKT. The authors find clear differences between the three cell lines utilized both in terms of fatty acid changes and the expression activation of the aforementioned signaling proteins. Overall, the study appears well designed and carefully carried out. This is a descriptive work, yet the results are clear and interesting and may be of pathophysiological relevance.

We thank very much this referee for the appreciation of our work. Here below are our point-by point answers to his/her comments.

  • Line 21. Given that the authors are referring to cells in this sentence, the word “synthesized” would seem more appropriate than “prepared”.

We changed the whole abstract and also part of the sentence.

  • Line 57. While the authors may be correct, it is also noteworthy that macrophages, which contain high levels of PUFA, are recently reported to contain significant amounts of sapienic acid (JLR 59: 237, 2018).

Thank you very much for this citation. We found very appropriate to cite this work that was also useful to discuss our results.

  • Line 120. The authors indicate that in Tables S7-9 the data are expressed in µg/ml. However in the captions to these tables, it is stated: “Values are expressed as relative quantitative percentages (% rel. quant.) calculated from the quantities of lipids obtained by HPLC analysis identified and calibrated by the standard references.” Please clarify. If the statement appearing in the captions is the correct one, it would be important for the authors to explain why this way of this presenting the data was chosen. The quantification method should also be described in sufficient detail. Please also specify whether CHO refers to total cholesterol or free cholesterol in these tables. Finally, the absence of data on other relevant phospholipid classes such as phosphatidylinositol or phosphatidylglycerol is conspicuous. Was there a technical reason to justify this absence?   

We thank the referee for this observation and apologize for being unprecise in our first version. In the new version we replaced at line 120  the notation g/mL with the one of relative quantitative percentages (% rel. quant.) as in the captures of tables; we apologize for this mistake due to a first version of the MS where we reported both Tables in g/mL and %rel. quant. We then chose to leave only the tables with the values expressed as % relative quantitative, i.e., obtained by calibration curves in µg/mL and converted to percentages of each lipid class over the total amount of lipid classes taken as 100%.Further, we followed the suggestion of the referee and added in SI details a new Section for the HPLC quantification procedures related to the lipid classes used as standards reference in this work. Specifically, for each lipid class we reported LOD and LOQ and R2 coming from the multiple points calibration curves, the repeatability of the method, expressed as RDS%( SD/mean*100) of  the retention times and peak area of specific  lipid class concentration, the inter-day variation results. We detailed that CHO indicates the content of free cholesterol in the sample. Regarding phosphatidylinositol or phosphatidylglycerol (PI and PG), we followed references for identification of lipid classes in breast cancer cells where the classes used in our work are reported to be 90% of the total lipids. We added these references (new refs 26 and 27) in the text to support our analysis. We added the notation regarding these two lipid classes and will take into account to add PI and PG among our standards for further work. 4)     Lines 189-193. I was very confused by these sentences. What I understood here is that the authors injected a 20-µl aliquot of a cell sample in distilled water, not a lipid extract, into the HPLC machine for lipid analysis. This makes little sense to me. Further, how does this fit with the following sentences, where the method of preparation of the lipid fraction from the remainder of the cell sample is properly described?  In line 191, I could not figure out which tables were the authors referring to.

We thank the reviewer for this comment and we rewrote the description of the experimental procedures.

As far as the preparation of the sample for HPLC injection is concerned (line 189-193 of the first version) we amended this paragraph to become clearer. The pellet sample (after breaking cells and centrifugation) was reconstituted in 1 mL of tridistilled water and 20 µL were injected into HPLC for the determination and quantification of phospholipid classes and free cholesterol. The rest was added with CHCl3:MeOH 2:1 in order to proceed with the other steps of lipid extraction followed by conversion to FAME by transesterification and GC analysis.  The reason to dissolve the crude pellet in H20 for the HPLC analysis and not after extraction, is due to technical and experimental conditions, since in HPLC (equipped with PEEK capillary tubing)  only the following aqueous miscible solvent can be used: MeOH, EtOH, 2-propanol (alcohols) and acetonitrile, while acetone (despite its aqueous miscibility), CHCl3,CHCl2, hexane, THF (common solvents used in NP/RP HPLC) for lipid characterization are absolutely not recommended because can cause swelling of the tubing and valves affecting the performance of the instrument. Even traces of chloroform invalidated our run also covering the UV signal. Since the mixture of eluents used in isocratic manner is 1M ammonium phosphate buffer/MeOH/2-propanol = 10:70:20, as described in Materials and Methods, we had repeatable results (as it can be seen in the Supporting Information looking at repeatability and inter-day variation) performing injection of 20 µL directly from the crude pellet dispersed in H20, also because the manual injector is continuously washed by the eluent mixture. Also, standard references were prepared in the same solvent mixture used for the runs. In SI we added representative HPLC run showing the sample vs the mix containing lipid standards (Figure S1).

The Tables cited in line 191 were a typo of the first version of the manuscript. We apologize for this mistake. The Tables referred to HPLC in Supporting Information are TABLES S7-9.

5) Line 197. The GC-FID analyses of fatty acids appear to have been conducted carefully and I have no concerns about these. The authors show good data on the phospholipid fatty acid profiles of their three cell lines under study (Tables 1-6). A fatty acid analysis of the different phospholipid subclasses separately (PC, PE, PS) would have also been very informative.

We agree with this comment. Nevertheless, our goal in this paper was to show the remodeling of the fatty acids induced by the supplementation and the immediate metabolism of sapienic acid, the latter being the most intriguing process – in our opinion – occurring to cancer cells. The detailed lipid classes composition and its role in the whole cancer cell behavior are objective of our future work with sapienic acid in cell cultures and exosomes.

6) Discussion – general. This section is too long and at times repetitive with Results. I suggest the authors to shorten this section and make it more focused on the results at hand.

The Discussion section has been reduced and optimized following the suggestions also of the second referee.

Reviewer 2 Report

Title: Sapienic acid metabolism influences membrane plasticity and protein signaling in breast cancer cell lines

With the importance of breast cancer, Ferreri et al. evaluated the influences of Sapienic acid on membrane plasticity and protein signaling. Overall, the whole structure of this study is good. However, some corrections are recommended for providing clear information. Particularly, I listed the following comments in detail here.

In the abstract, the background is too long. The author needs to mention the aim of the study at the end of the background. Methods should be added, also, the finding of the assay could be added step by step based on methods. All of the terms should be completely mentioned for the first time, for example, PUFA, EGFR and so on.

In the introduction. Please add data on the importance and mortality of cancer in particular breast cancer in the world. The references are missed for some sentences, for example: “On the other hand, since eukaryotic cells, and among them cancer cells, cannot be formed without PUFA, the two families of n-6 and n-3 fatty acids are needed, starting from the dietary intakes of essential fatty acids (EFA) precursors, linoleic and alpha-linolenic acids, respectively, (Scheme 2). More recently, palmitic acid was found to be involved by an unusual pathway of delta-6 desaturation and transformed into sapienic acid (SA) (Scheme 1).” and so on.

In Materials and Methods:

Why the 50 µM concentration was used for tests? Do you have rationale reason or document for this concentration? 

All of the terms should be completely mentioned for the first time, for example, MTT, FAME and so on.

Please add references of protocols for tests and assays.

In line 155, change “doses” to concentrations.

In discussion, add references for some sentences. For example, “Research on the connections between fatty acid composition of membrane phospholipids and cancer proliferation has attracted a lot of attention in the last decades and, in particular, the follow-up of the EGFR/AKT/mTOR pathway showed its ubiquity in tumors to maintain growth, survival, and metastasis.”, and so on. Also, discuss your results before relating them to the results of other published work. Precise conclusion as it’s too short in its current form. Add a significant statement that must be structured as “what was offered by authors? Do the authors have more thoughts on this field?

Author Response

REFEREE 2

Title: Sapienic acid metabolism influences membrane plasticity and protein signaling in breast cancer cell lines

With the importance of breast cancer, Ferreri et al. evaluated the influences of Sapienic acid on membrane plasticity and protein signaling. Overall, the whole structure of this study is good. However, some corrections are recommended for providing clear information. Particularly, I listed the following comments in detail here.

Thank you for this introductory comment. We proceeded with the corrections requested by this referee and report point-by-point the answers below.

In the abstract, the background is too long. The author needs to mention the aim of the study at the end of the background. Methods should be added, also, the finding of the assay could be added step by step based on methods. All of the terms should be completely mentioned for the first time, for example, PUFA, EGFR and so on.

We followed this suggestion and revised the abstract as well as the abbreviations

In the introduction. Please add data on the importance and mortality of cancer in particular breast cancer in the world. The references are missed for some sentences, for example: “On the other hand, since eukaryotic cells, and among them cancer cells, cannot be formed without PUFA, the two families of n-6 and n-3 fatty acids are needed, starting from the dietary intakes of essential fatty acids (EFA) precursors, linoleic and alpha-linolenic acids, respectively, (Scheme 2). More recently, palmitic acid was found to be involved by an unusual pathway of delta-6 desaturation and transformed into sapienic acid (SA) (Scheme 1).” and so on.

We added the new ref 7 referred to the last statistics GLOBOSCAN for breast cancer and used already cited references to highlight the other points of the introduction.

In Materials and Methods:

Why the 50 µM concentration was used for tests? Do you have rationale reason or document for this concentration? 

We thank the reviewer to have raised this question that allows us to detail better this choice.

As written in the first version, we determined the IC50 of SA in three breast cancer cell lines and mentioned that we chose this concentration since it is not toxic. In literature, effects of sapienic acid were reported at 10 and 25 µM concentrations (see ref 13) in normal cells (phagocytic cells), actually indicating that only at 25 µM there was an anti-inflammatory effect. We decided to use the 50 µM concentration taking into account that our experiment is in cancer cells and we needed to simulate a larger production of this compound in cancer cells (delta-6 desaturase activation), which is still comfortable with the IC50 (2-3 times more than 50 µM for all cell types tested in this work). Moreover, we evidenced that no lipid accumulation (formation of triglycerides) is obtained at this concentration and transformation into its n-10 congeners as well as incorporation in membrane lipids occurs. In the new version we added such thorough explanation in the Discussion.

All of the terms should be completely mentioned for the first time, for example, MTT, FAME and so on.

We took care of this comment

Please add references of protocols for tests and assays.

Added new refs 22-24 for MTT test and also for the other tests

In line 155, change “doses” to concentrations.

We changed as suggested

In discussion, add references for some sentences. For example, “Research on the connections between fatty acid composition of membrane phospholipids and cancer proliferation has attracted a lot of attention in the last decades and, in particular, the follow-up of the EGFR/AKT/mTOR pathway showed its ubiquity in tumors to maintain growth, survival, and metastasis.”, and so on. Also, discuss your results before relating them to the results of other published work. Precise conclusion as it’s too short in its current form. Add a significant statement that must be structured as “what was offered by authors? Do the authors have more thoughts on this field?

We amended the Discussion as suggested by the referee. We added references for the mentioned paragraph (new refs 34 and 35). We reorganized the references according to the text reorganization. We extended Conclusions as requested especially expressing better our aim to use our work in cells to understand better lipid metabolism and biomarkers useful in translational studies of breast cancer patients.

We must also be cautious at this point to extend our thoughts beyond the experimentally proved results, however we believe that we demonstrated that the accurate follow-up of membrane fatty acids still produces new insights in cancer fatty acids management, translationally useful to nutritional and metabolic follow up of patients.

Round 2

Reviewer 1 Report

None.

Reviewer 2 Report

I recommend acceptance of the current version.